# Homologous Recombination Repair in Biliary Tract Cancers: A Prime Target for PARP Inhibition?

**DOI:** 10.3390/cancers14102561

**Published:** 2022-05-23

**Authors:** Chao Yin, Monika Kulasekaran, Tina Roy, Brennan Decker, Sonja Alexander, Mathew Margolis, Reena C. Jha, Gary M. Kupfer, Aiwu R. He

**Affiliations:** 1Ruesch Center for the Cure of Gastrointestinal Cancers, Lombardi Comprehensive Cancer Center, Georgetown University, Washington, DC 20007, USA; chao.yin@gunet.georgetown.edu (C.Y.); monika.kulasekaran@gunet.georgetown.edu (M.K.); tina.roy@medstar.net (T.R.); 2Foundation Medicine, Cambridge, MA 20007, USA; bdecker@foundationmedicine.onmicrosoft.com (B.D.); salexander@foundationmedicine.com (S.A.); mmargolis@foundationmedicine.com (M.M.); 3Department of Radiology, Georgetown University Medical Center, Washington, DC 20007, USA; jhar@gunet.georgetown.edu; 4Departments of Oncology and Pediatrics, Lombardi Comprehensive Cancer Center, Georgetown University, Washington, DC 20007, USA; gary.kupfer@georgetown.edu

**Keywords:** homologous recombination repair, DNA damage repair, PARP inhibitor, biliary tract cancer, synthetic lethality

## Abstract

**Simple Summary:**

Biliary tract cancers (BTCs) are a rare but deadly group of gastrointestinal tumors that are often diagnosed in the advanced stages of disease. Despite large studies investigating optimal systemic therapy options in BTCs, current chemotherapies provide only modest benefits in overall survival. The rapidly evolving study of homologous recombination repair (HRR) as part of the broader DNA damage repair (DDR) system makes it possible to exploit deficiencies in this pathway with targeted agents such as PARP inhibitors (PARPi). We will review the rationale for PARPi use, as well as considerations for further unlocking their potential in treating BTC.

**Abstract:**

Biliary tract cancers (BTCs) are a heterogeneous group of malignancies that make up ~7% of all gastrointestinal tumors. It is notably aggressive and difficult to treat; in fact, >70% of patients with BTC are diagnosed at an advanced, unresectable stage and are not amenable to curative therapy. For these patients, chemotherapy has been the mainstay treatment, providing an inadequate overall survival of less than one year. Despite the boom in targeted therapies over the past decade, only a few targeted agents have been approved in BTCs (i.e., IDH1 and FGFR inhibitors), perhaps in part due to its relatively low incidence. This review will explore current data on PARP inhibitors (PARPi) used in homologous recombination deficiency (HRD), particularly with respect to BTCs. Greater than 28% of BTC cases harbor mutations in genes involved in homologous recombination repair (HRR). We will summarize the mechanisms for PARPi and its role in synthetic lethality and describe select genes in the HRR pathway contributing to HRD. We will provide our rationale for expanding patient eligibility for PARPi use based on literature and anecdotal evidence pertaining to mutations in HRR genes, such as *RAD*51C, and the potential use of reliable surrogate markers of HRD.

## 1. Introduction

Biliary tract cancers (BTCs) arise from bile duct epithelia within the liver (intrahepatic) and outside the liver (extrahepatic). They are heterogeneous and constitute ~7% of all gastrointestinal (GI) tumors [1]. In the Western world, BTC occurs at a rate of 0.5 to 2 cases per 100,000 people [2]. BTCs tend to be aggressive, with a 5-year OS of <10% [3]. Surgery is the only means of a cure; however, upwards of 70% of patients are diagnosed with advanced disease that is unresectable [2,4,5]. Studies found that patient median disease-free survival (DFS) was 12–36 months after surgical resection, and their overall survival (OS) was high as 80 months after R0 resection [6,7,8]. Still, disease recurrence after curative surgery remains an issue, and recurrence rates are as high as 40–70% [9,10,11].

Until the last couple of years, the standard treatment for BTCs was chemotherapy with palliative intent. Gemcitabine plus cisplatin (GEMCIS) is still the most commonly used front-line regimen in the United States, and fluorouracil plus oxaliplatin (FOLFOX) is the typical second-line treatment, supported by the Phase III ABC-02 trial and the ABC-06 trial, respectively [12,13]. Nonetheless, survival of patients with advanced BTCs remains abysmal, with mOS being under one year with traditional chemotherapy [14].

Over the past decade, breakthrough therapies in BTC have revolved around targeted therapies and immunotherapies. Increased use of broad molecular profiling has led to the finding that BTCs have some of the highest genetic aberration rates of all cancers. Studies have shown that upwards of 60% of cholangiocarcinoma (CCA) contain such aberrations, including those in isocitrate dehydrogenase-1 (*IDH*1), fibroblast growth factor receptor-2 (*FGFR*2), and neurotrophic tyrosine receptor kinase (*NTRK*) [15,16,17,18]. Accordingly, molecularly targeted agents have been developed and studied in BTC, leading to the recent United States Food and Drug Administration (USFDA) approval of ivosidenib (IDH1 inhibitor) and pemigatinib (FGFR 1,2,3 inhibitors) [19,20].

Similar to the mutations named above, the focus of our review is on the role of poly (ADP-ribose) polymerase inhibitors (PARPi) in homologous recombination repair (HRR) deficient tumors, particularly concerning BTCs. This mainly encompasses genetic alterations that are linked to genes in the HR pathway, including but not limited to *ARID1A, ATM*, *ATRX*, *BAP1*, *BARD1*, *BLM*, *BRCA1/2*, *BRIP1*, *CHEK1/2*, *FANCA/C/D2/E/F/G/L*, *MRE11A*, *NBN*, *PALB2*, *RAD50*, *RAD51*, or *WRN*. Retrospective gene analysis found evidence of HRR deficiency (HRD) in 28.9% of BTCs, one of the highest HRD rates of all cancers [21,22,23]. Figure 1 presents the relative frequency data for these gene mutations in BTC, extracted from the COSMIC/TCGA database. PARP inhibitors (PARPi) are the first group of drugs designed to exploit cancer HRD. Preclinical studies suggest that BTC is susceptible to PARP inhibition. Several PARPi, including olaparib, rucaparib, and niraparib, are under clinical investigation in BTCs [24]. In this review, we will discuss the predictors of PARPi sensitivity and resistance in both clinical and preclinical studies. Furthermore, synergistic strategies through combination therapy with other drugs will be addressed.

## 2. The DNA Damage Repair Pathway

Normal cells are under constant stress from external and internal factors, which can cause cell and DNA damage. Cells rely on intricate DNA damage repair (DDR) pathways to identify and fix errors to ensure genomic integrity. Various DNA repair mechanisms exist to address an array of different mutations and alterations. The major DNA repair pathways include (but are not limited to) base excision repair (BER), nucleotide excision repair (NER), mismatch repair (MMR), non-homologous end joining (NHEJ), and homologous recombination repair (HRR). For example, BER has activity in repairing single-stranded breaks (SSB), while double-stranded breaks (DSB), often highly cytotoxic if not repaired correctly, are typically fixed through error-proof HRR or more error-prone classical NHEJ and alternative-NHEJ pathways such as microhomology-mediated end joining (MMEJ) [25,26]. Specifically, HRR is active during the S and G2 phase of the cell cycle using a sister chromatid template for DSB repair and is also involved in replication forks that have stalled or collapsed [26,27].

In response to DNA damage, these main DDR routes operate under a dynamic and complex network of overlapping signaling pathways that crosstalk and either promote genomic stability or trigger programmed cell death, as deemed appropriate under the circumstances [28]. DDR defects theoretically provide a number of advantages during carcinogenesis; an increased mutation rate helps tumors withstand replicative stress and offers an adaptive advantage. Mutations in DDR genes have been identified in many cancers, and it has been proposed that every DDR mechanism is affected during tumorigenesis [29,30]. However, the impact of many genes in the DDR pathway is not yet well understood. Herein, we will discuss the evidence surrounding the *BRCA*-mutant phenotype in some *BRCA*-wild type tumors and the need for additional biomarkers to detect this phenotype. Table 1 summarizes a non-exhaustive list of HRD-related genes, their frequencies in CCA, and the evidence surrounding their sensitivity to DDR-targeting agents.

## 3. Biomarkers of HRD and Predictors of PARPi Response in BTC

PARPi are the first and only class of drugs approved clinically to target DDR deficiency. PARP proteins are essential to DNA repair in a compensatory pathway to HRR that facilitates the repair of SSBs. PARPi have cytotoxic properties via two mechanisms: (1) they inhibit the enzymatic activity of PARP1 and PARP2 by binding to their cofactor β-NDA+ binding site, and (2) they “trap” PARP1 and thereby prevent the release of PARP1 from damaged DNA, leading to cell death in dsDNA repair-deficient cells [73]. PARP1 was initially described as a key facilitator of BER in ssDNA repair, although there is now increasing evidence that it also has direct roles in non-HRR pathways such as NHEJ [74,75,76]. PARPi efficacy in cancers with HRD exploits the concept of synthetic lethality. Synthetic lethality occurs when two (or more) DDR pathways are blocked, preventing effective DNA repair [74]. For example, if a *BRCA* mutation renders HRR defective, pharmacologic inhibition of PARP enables synthetic lethality by blocking at least two other repair pathways.

Although PARPi were initially studied and approved in *BRCA*1/2 mutated ovarian cancers, its use has been expanded to other cancers, including breast cancer, pancreatic cancer, and prostate cancer. Currently, not a single PARPi has been approved by the USFDA for BTC, although several trials are investigating PARPi use in BTC, and published data are pending. Therefore, much of this discussion is supported by data extrapolated from other cancers and preclinical models while being cautious of the biological differences between different tumor types.

When considering predictors of response to PARPi, *BRCA* 1 and 2 represent the iconic gene mutations that disrupt HRR and increase the sensitivity of cancer cells to PARPi. The phase III PAOLA trial impressively reported improved mOS of 37.2 months vs. 17.7 months (HR 0.33, 95% CI 0.25–0.45), with the addition of olaparib to bevacizumab vs. bevacizumab alone in stage III/IV *BRCA* mutated ovarian cancer in the first-line maintenance setting [61]. However, the investigators noted that in patients with HRD-positive tumors without *BRCA* mutations, the mOS was still much improved with the addition of olaparib (28.1 months vs. 15.5 months, HR 0.43, CI 95% 0.28–0.66). The SOLO3 trial showed that treatment with olaparib in platinum-sensitive, recurrent *BRCA*-positive ovarian cancer yielded an objective response rate (ORR) of 72% vs. 51% with nonplatinum chemotherapy [77].

The DDR pathway is complex; however, it potentially bears many opportunities for targeted therapies. In fact, as previously alluded to, varying degrees of *BRCA*-mutated phenotype, or “*BRCA*ness”, are evident across many cancers that are *BRCA*-wild type or have *BRCA* mutations of unclear significance. With particular relevance to BTC, we propose two goals for this review: (1) exploring the qualifications for patients who may benefit from DDR targeting drugs such as PARPi beyond just *BRCA* mutations and (2) exploring the mechanisms of PARPi refractoriness, resistance, ability, and potential ways to overcome this.

As mentioned earlier, the PAOLA trial for ovarian cancer demonstrated significant benefit from olaparib for patients who were HRD-positive and *BRCA*-negative. In BTC, while only 3–4% of tumors cary BRCA mutations, as high as 28.9% of tumors carry mutations in genes involved in the HRR pathway [21,78]. Hence, we seek to characterize a group of patients with BRCA wild type BTC who may benefit from DDR targeting therapies such as PARPi.

## 4. *ARID*1A

To explore a few common DDR mutations in BTC, the most commonly detected gene alteration is *ARID*1A (AT-rich interactive domain-containing protein 1A), occurring in 13–15% of BTC samples [21,22,23]. Most *ARID*1A mutations are truncating mutations (92%), leading to the gene’s inactivation and loss of its protein expression [79]. The clinicopathologic features of *ARID*1A are somewhat controversial. Most studies suggest a tumor suppressor role for this gene, whereas others describe pro-oncogenic properties [79]. Its oncogenic properties are not well understood and are seemingly complex; *ARID*1A appears to increase oxidative stress, such as by increasing CYP450 in the liver in hepatocellular carcinoma (HCC). It is overexpressed in primary HCC but, interestingly, lost in metastatic HCC [80].

As the most frequently mutated chromatin regulator across all cancers, it is important to note the role of *ARID*1A in global transcription regulation [81]. Specifically, depletion of *ARID*1A leads to decreased RNA polymerase II (RNAPII) pausing. The pausing of RNAPII is essential for maintaining cell homeostasis in response to environmental stimuli and limiting excessive transcription [81]. Furthermore, co-occurring TP53 and *ARID*1A mutations are rare; this mutual exclusivity has been demonstrated in endometrial, gastric, breast, and esophageal cancers [82]. Studies found that some, but not all, transcriptional dysregulation suffered from *ARID*1A loss can be rescued by *ARID*1B and TP53 targets [81]. In fact, the rare coexistence of TP53 and *ARID*1A mutations often leads to more aggressive cancers [82].

When considering its role in DDR, *ARID*1A exerts its role in both HR and NHEJ pathways. It is also required for proper G2/M DNA damage checkpoint, which may lead to insufficient cell cycle arrest to repair DSB [31]. Its deficiency has been demonstrated to impair the DSB end resection necessary for repair. *ARID*1A is a member of the SWI/SNF complex and is involved in chromatin remodeling, which promotes efficient DSB end resection and sustains ATP-dependent signaling [31,32]. The aberrancies in the SWI/SNF complex as a whole have been linked to various cancers and linked to tumor suppression. Interestingly, *ARID*1A and *TP53* are often found to be co-mutated in CCA, which would indicate a more aggressive cancer given the collaboration of these two genes in the activation of downstream effectors to prevent tumorigenesis, as noted above [83,84].

In preclinical studies, cell lines with knockout *ARID*1A demonstrated sensitivity to PARPi. Moreover, selective inhibition of *ARID*1A-deficient xenografts by PARPi was observed [31]. These findings have not been validated clinically. On the contrary, retrospective analysis of data from the ARIEL2 trial identified *ARID*1A persistence as a factor of rucaparib resistance. He and colleagues found that all 10 ovarian cancer patients with loss of ARID1A and BRCA-wild type from the study had significantly less PFS on rucaparib than proficient *ARID*1A cancers [34]. Nonetheless, there is insufficient data to delineate the mechanism of such resistance or suggest the clinical prognostic or predictive role of ARID1A in BTC. Further prospective clinical trials are necessary to evaluate PARPi efficacy in *ARID*1A mutated cancers, particularly given its relatively high frequency across all tumor types.

## 5. *ATM* and *ATR*

*ATM* (Ataxia-telangiectasia mutated) and *ATR* (*ATM*- and Rad3-related) are part of the PI3K-related kinase family. When mutated, they are associated with complex syndromes that exhibit nervous system phenotypes and DDR deficiencies [85]. *ATM* and *ATR* mutations are also among the most common associated HRD mutations in BTC, occurring in 5.7% and 5.1% of BTC tumor samples, respectively [22,23]. ATM functions upstream in the HRR pathway, and both help phosphorylate downstream HRR effectors, including BRCA1. ATM and ATR show complementary relationships, likely through compensatory mechanisms, such that *ATM* knockout cells have increased levels of *ATR* [85]. Hence, there is the suggestion that the two genes work together in the DDR pathway, and it would be reasonable to consider this as a mechanism of resistance to DDR targeting agents in *ATM* or *ATR* deficient cells.

It has been shown that patients with *ATM*-deficient tumors are more responsive to radiotherapy, platinum-based chemotherapy, and PARPi [35]. Similarly, in ovarian cancer xenografts, a group showed that administering an ATR inhibitor helped overcome platinum resistance [41]. A phase II double-blind trial of olaparib plus paclitaxel vs. paclitaxel alone showed significant improvement in mOS (13.1 v 8.3 months, HR 0.35, *p* = 0.002) for patients with recurrent or metastatic gastric cancer with low *ATM* expression on immunostaining [86]. Given this clinical success, efforts are made to study ATM inhibitors, which artificially block normal ATM function. Several ATM inhibitors have entered early clinical phases in combination with cytotoxic therapies, PARPi, and/or radiation [NCT02588105, NCT03423628, NCT03571438].

## 6. IDH1

*IDH1* mutations have a frequency of 20–25% in intrahepatic CCA. They indicate that *BRCA*ness may not be exclusive to genes within the DDR pathway. Mutations in IDH1 across most tumors are predominantly somatic rather than germline mutations. They are always heterozygous, primarily because of the driver R132H mutation, consistent with gain of function and dominance over the remaining wild-type allele [87]. This mutant gene converts the α-ketoglutarate to 2-hydroxyglutarate (2-HG). 2-HG directly inhibits HRR, which was well captured by Sulkowski and colleagues [88]. In HCT116 and HeLa cell lines, these investigators first confirmed the existence of mutant IDH1 protein expression and a 100-fold increase in 2HG production. Compared to the IDH1-wild-type cell line, HCT116 and HeLa cell lines had a markedly reduced capacity for DSB repair after ionizing radiation exposure. Further testing via a plasmid reporter assay to compare relative DSB repair activity showed a marked deficiency in HRR in the mutant cells [88]. These mutational changes also sensitized the cell lines to further attack by PARPi. Although the exact mechanisms of its involvement in the HRR remain to be elucidated, mutant IDH1 was at least found to independently downregulate *ATM* in a mouse model [89].

In IDH1 mutated cancers, including BTC, PARP inhibitor efficacy is evident in preclinical studies but requires further clinical testing. IDH1 inhibitor ivosidenib has been approved by USFDA for CCA after demonstrating significantly improved PFS (6.9 months vs. 2.7 months in the placebo arm, *p* < 0.001) in a phase III trial. Although clinical data on PARPi use in IDH1 mutated CCA is lacking, olaparib is currently being tested in patients with advanced glioma or cholangiocarcinoma with IDH1/2 mutations [NCT03212274]. A phase II study of olaparib in IDH1/IDH2-mutant mesenchymal sarcoma also reported some signs of clinical benefit [90].

## 7. *RAD*52

Among several roles in DNA repair and replication, *RAD*52 binds single-stranded DNA and plays key roles in single-strand annealing and HRR of DSBs. In mammals, *RAD*52 has diminished HRR compared to that of other proteins, including *BRCA*1/2. In checkpoint-deficient cells, it facilitates break-induced replication (BIR), which is a specialized pathway that repairs single-ended DBS and promotes the finalization of DNA replication [66,67,68]. Interestingly, in the study of gene mutations across different malignancies, a *RAD*52 mutation was found to sensitize *BRCA*-deficient AML cells to PARPi. Deficiencies in KU70/80 proteins that compete with *RAD*52 in binding to DSB are related to resistance to PARPi, although mutations in KU70/80 have not been reported in BTC [91,92].

One of its most prominent functions is the formation of a Rad51–Rad52–Rad59 complex, which is involved in conservative recombination events, including gene conversion and reciprocal recombination [67]. *RAD*52-mediated DNA repair remains active in PARPi-treated *BRCA*-deficient tumor cells such that dual inhibition of *RAD*52 and PARP1 have demonstrated evidence of synthetic lethality in a *BRCA*-deficient mouse model [69]. It has other important functions in DNA repair, which are beyond the scope of the article.

## 8. *RAD*51 and *RAD*51C

The HRR genes of particular interest are *RAD*51 and its paralogs (e.g., *RAD*51C). During DSB end resection, 3′-tailed ends are created in the DNA and are quickly bound by replication protein A (RPA). RPA is then displaced by *RAD*51, which is central to a helical nucleoprotein filament that identifies and invades a homologous donor DNA duplex [93]. *RAD*51 is helped by other HRR proteins, most prominently BRCA2, which is recruited by BRCA1 and co-localizes with PALB2 at the site of DSB. When bound to RAD51, BRCA2 attenuates ATPase activity in RAD51 and assists in RAD51 filament formation and stabilization of the complex [93]. From there, RAD51 mediates homologous DNA strand identification and strand invasion to begin DNA replication in the recombination process.

Clinically, the correlation between *RAD*51 expression and tumorigenesis is complex. For example, low *RAD*51 expression correlated with high histologic grade in breast cancer but was predictive of higher complete pathologic response rates to neoadjuvant chemotherapy [94]. However, evidence also suggests that *RAD*51 expression is increased during breast cancer progression, and overexpression of *RAD*51 in colorectal cancer was a predictor of poor outcome [95,96]. During therapy with PARPi, immunostaining showed that increased expression of *RAD*51 nuclear foci correlated with PARPi resistance in *BRCA* mutated tumors [64]. Simultaneous targeting of *RAD*51 and PARP inhibition has been proposed as a synergistic approach, although there is currently no clinical data to confirm this proposal [94,96].

There are five paralogs of *RAD*51, including *RAD*51B, *RAD*51C, and *RAD*51D, which share similar functions in HRR. *RAD*51C mutations occur in 1.8% of BTC [22,23]. *RAD*51C colocalizes with *RAD*51 to DNA repair foci and also facilitates nucleoprotein filament stabilization. It participates in the formation of two complexes: the XRCC3-*RAD*51C (CX3) heterodimeric structure and *RAD*51B-*RAD*51C-*RAD*51D-XRCC2 (BCDX2) heterotetrameric structure [97]. Compared to the other *RAD*51 paralogs, this suggests that *RAD*51C plays a more prominent role in HRR given its unique involvement in both structures, which are central to the early and late stages of HR [98]. The CX3 and the BCDX2 complexes are also crucial for stalled replication fork reversal and efficient restart, for which HRR components are also generally implicated [99].

Germline *RAD*51C pathologic variants have been linked with an increased risk of tubo-ovarian carcinoma (*RAD*51C mutant RR 7.55, 95% CI = 5.60–10.19) and breast cancer (*RAD*51C mutant RR 1.99, 95% CI = 1.39–2.85) [100]. Clinically, immunostaining of high-grade serous ovarian cancer showed higher levels of *RAD*51C protein compared to benign tumors, and increased *RAD*51C levels were associated with higher clinical staging and poorer prognosis [97]. Given its integral role in HRR, studies suggest that *RAD*51C shares many tumor characteristics associated with *BRCA* germline variants [101]. In fact, in the ARIEL2 trial of the PARP inhibitor rucaparib in ovarian cancer, tumor biopsies revealed positive associations between alterations in *BRCA*1 or *RAD*51C, high genomic loss of heterozygosity (gLOH), and increased response to rucaparib [102]. Silencing of *RAD*51C expression has also been shown to make tumor cells more sensitive to PARPi in vitro and has also been replicated in clinical trials for patients with ovarian cancer who had an increased response rate to rucaparib [101,103]. Conversely, secondary mutations in *RAD*51C, which restore the function of *RAD*51C, are linked to the restoration of the HRR pathway and have been identified as a mechanism of PARPi resistance in xenograft models [104].

## 9. Genomic Loss of Heterozygosity

The last *BRCA*ness marker we would like to discuss is the concept of gLOH, which is a measure of allelic imbalance when heterozygous somatic cells become homozygous due to the loss of one of the two alleles. In a retrospective study, biallelic *BRCA*1/2 alteration was associated with increased LoH across multiple cancer types, including biliary tract cancers (BTC), although the magnitude of this association was variable across each cancer type [105]. In BTC, the odds ratio for this association was 21.5. By plotting the sensitivity and specificity of classifying *BRCA*1/2 biallelic alteration vs. wild type using genomic LoH, the authors found a cutoff of >17.6% genomic LoH for BTC, which is very close to the >16% genomic LoH which was used in the ARIEL trial of rucaparib in ovarian cancer [106]. Although >17.6% may correspond to a *BRCA*-mutant phenotype, prospective clinical trials are necessary to evaluate varying levels of genomic LoH, including those with genomic LoH >17.6%, and the corresponding sensitivity to DDR-directed therapies, such as PARPi. The goal would be to find a gLOH cutoff representing a level of HRR deficiency that would sensitize BTC patients to such therapies in the hope that gLOH could become an umbrella biomarker that catches all clinical HRR deficient cancers.

## 10. Anecdotal Case of *RAD*51 Mutation in CCA

We describe our institutional experience of a 73-year-old woman presenting with metastatic BTC in March 2019. She was treated with GEMCIS between March and August 2019 and was switched to FOLFOX and fluorouracil maintenance therapy upon disease progression from August 2019 to December 2020, when she had further disease progression. She was then enrolled in our Phase II trial of olaparib plus pembrolizumab in relapsed/refractory advanced BTC in February 2021. As anticipated from her good response to platinum-based chemotherapy, the patient had an excellent partial response to PARPi in addition to ICI and a 65% decrease in index lesions (Figure 2). FoundationOne CDx comprehensive genomic profiling of the tumor showed a t(13;17) translocation with the chromosome 17 breakpoint identified in exon 4 of *RAD*51C (Figure 3A). This rearrangement had strong bi-directional evidence, with 237 supporting reads. Examination of chromosome 17 copy number data shows a copy number transition and accompanying LOH with a breakpoint at the *RAD*51C locus, suggesting likely loss of the second allele (Figure 3B). Genomic LOH, an orthogonal HRD signature validated as a companion diagnostic for rucaparib in ovarian cancer, was 11%, which is below the 16% threshold for gLOH-high status set in ovarian cancer. Biallelic inactivation of *RAD*51C has been associated with elevated gLOH [107]. However, the observed distribution of gLOH values varies by cancer type, and further study is needed to identify clinically relevant disease-specific thresholds [105]. The patient’s treatment was complicated by grade 3 immune hepatitis in June 2021; pembrolizumab has been held since then. She continued to respond to olaparib alone at the most recent clinical evaluation in February 2022.

## 11. Augmenting PARPi Efficacy

So far, we have explored aberrations in the HRR pathway that may suggest increased sensitivity to PARPi outside of the well-studied *BRCA*1/2 mutations. However, we turn our attention to ways to augment the efficacy of PARP inhibitors in BTC and other cancers.

First, we want to touch on the concept of drug resistance briefly. Tumor cells have the ability to develop ways to disrupt the availability of PARPi within the cell. For example, a high expression of ABCB1a/b, a drug-efflux transporter gene, has been associated with resistance to PARPi in mouse models [108]. ABCB1a/b overexpression is also a mechanism of resistance to topoisomerase inhibitors. However, by blocking the MDR1 glycoprotein encoded by ABCB1, researchers have demonstrated re-sensitization of ovarian cancer cell lines to PARPi. The co-administration of PARPi with MDR1 inhibitors, such as tariquidar and verapamil, or the development of PARPi that are non-substrates for MDR1-mediated efflux, is being studied [108].

The restoration of the HRR pathway in HRR-deficient cancers appears to occur by several different mechanisms. While this is sometimes achieved via compensatory mechanisms, such as increased activity of *RAD*52, studies have also demonstrated secondary somatic reversion of mutated genes in the DDR pathway, such as in *BRCA*1/2 and *RAD*51 genes, which can restore their function in cancer cells [68,109]. In high-grade ovarian cancers with germline *BRCA*1/2 mutations, almost 50% of platinum-resistant tumors demonstrated secondary mutations in the *BRCA*1/2 gene [109]. Several drug combination approaches have been proposed in these scenarios. For example, there is interest in CDK inhibition, given that CDK1 has a direct impact on *BRCA*1/2 activation through phosphorylation of *BRCA*1/2 [110]. PI3K inhibitors can also downregulate *BRCA* expression in cell lines and human-derived xenografts, and this downregulation is likely mediated by ERK signaling and impaired recruitment of *RAD*51 to DNA repair sites [110,111]. An early phase I clinical trial suggested synergy between alpelisib (PI3K inhibitor) and olaparib in ovarian cancer [112].

The idea of adding secondary agents to combat PARPi resistance or increase efficacy segues into another fascinating area of drug synergy: PARP plus PD-1/PD-L1 inhibition. Data indicate that *BRCA*1/2 deficient cancers express higher levels of neoantigens, thereby making themselves more immunogenic. The DNA damage created by PARP inhibitors generates an interferon response that leads to increased T-cell recruitment and TILs [113]. For example, preclinical studies demonstrate synergy between PARP inhibition and anti-CTLA-4 therapy in *BRCA*1/2 mutant ovarian cancer [114]. An interaction between PARP inhibitor and tumor-associated immunosuppression likely provides evidence to support the combination of PARP inhibitors and anti-PD-1/PD-L1 combinations. PARP inhibitor-related upregulation of PD-L1 expression in breast cancer cell lines and animal models appears to occur by knocking out GSK3β activity, which significantly increases PD-L1 expression and resistance to PARP inhibition. Hence, the blockade of PD-L1 re-sensitized tumor cells to PARP inhibition [115]. In an ongoing Phase II study at our institution, we are studying the combination of pembrolizumab and olaparib as subsequent-line therapy in BTC (NCT04306367) [116]. A separate Phase II study is looking at combined durvalumab plus olaparib in IDH-mutated solid tumors, including a cohort of patients with BTC (NCT03991832) [117].

Even more pertinent to the HRR pathway, several inhibitors of specific HRR genes, such as ATM, are now under preclinical development and early clinical investigation. A common practice in studying isolated genes in basic and translational research is to artificially knock out the gene of interest and observe downstream effects. What if we artificially induce DDR in tumor cells to increase their susceptibility to synthetic lethality with PARPi? Current ATM inhibitors include KU-60019 and AZD0156, which, when combined with PARPi in HRR stable cell lines, induce a massive increase in DNA damage and dysregulate the G2 DNA damage checkpoint [118]. This synergy and checkpoint dysregulation is also seen when a CHK1 inhibitor (PF-477736) was combined with rucaparib in cell lines [119].

## 12. Conclusions

Many factors limit the advancement of therapeutics in BTC, including anatomical challenges prohibiting ready access to tissue biopsy, locally advanced disease precluding curative resection, and the complicated biology of this aggressive cancer. Thus, there is an urgent need to develop new strategies to anticipate a BTC diagnosis at an early and resectable stage and obtain sufficient tissue samples to perform genomic analysis. Still, these limitations and needs should not distract from our recent achievements, including the incorporation of targeted agents and immunotherapies into our treatment arsenal and their use in place of traditional chemotherapy. Moreover, the mechanisms and intricacies of the HRR pathway continue to be elucidated. There is increasing evidence that the *BRCA* phenotype leading to HRD is not exclusive to *BRCA*1/2 mutations but rather encompasses many more genes in the HRR pathway. However, we recognize that each of the genes involved in HRR may not have equal significance, and there are varying degrees of preclinical and clinical evidence for the degree of contribution of each to *BRCA*ness [120].

Our review is timely due to a better understanding of HRR, advancements in molecular profiling of cancers, and the availability of therapeutic agents targeting HRD cancer. Particularly in BTC, harboring one of the highest frequencies of HRR aberrations, the suboptimal standard chemotherapy represents an urgent call for newer approaches to cancer therapy. PARPi is the first class of drugs used clinically to target HRD cancers via synthetic lethality and has been well studied in *BRCA*1/2 mutated cancers, including pancreas, prostate, breast, and ovarian, generally with very positive results. In BTC, *BRCA*-mutant tumors occur at a frequency of <5%, but aberrations in HRR genes overall make up >25% of cases [22,23]. Although most data are currently limited to animal models and cell lines, we explored evidence supporting a role for several more frequently mutated HRR genes and their possible mechanisms of action in creating BRCAness in BTC. Ultimately, we encourage prospective studies that redefine and expand the qualification of patients for DDR-targeting drugs such as PARPi and investigate strategies to augment the effectiveness of these inhibitors.

## Figures and Tables

**Figure 1 cancers-14-02561-f001:**
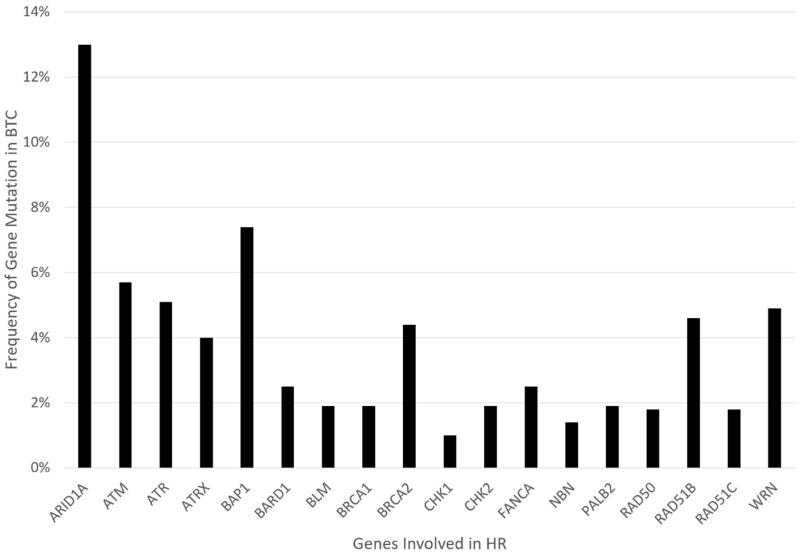
Graphical representation of mutation frequency in genes involved in HR.

**Figure 2 cancers-14-02561-f002:**
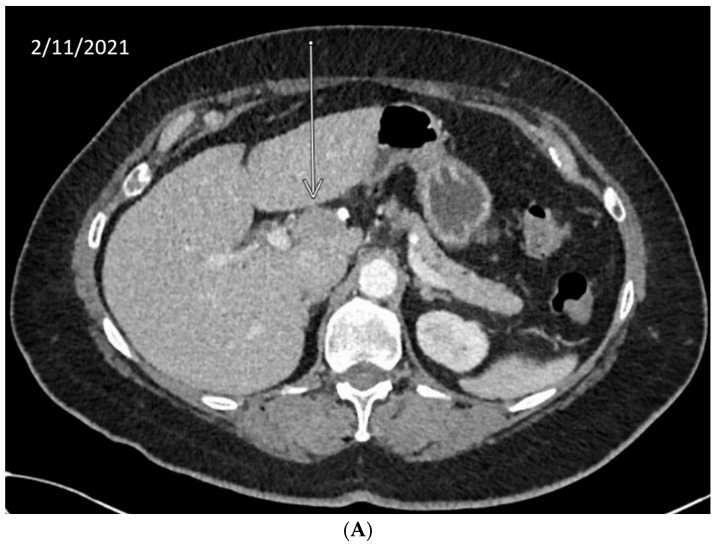
Time-lapsed CT images of a patient with intrahepatic CCA and *RAD*51C mutation (via t(13;17) translocation with the chromosome 17 breakpoint identified in intron 4 of *RAD*51C). She concurrently had a genomic loss of heterozygosity (gLOH) of 11%. The patient had an excellent response to treatment with PARPi and ICI and a 65% decrease in index lesion (shown by arrow). (**A**) CT from February 11, 2021. (**B**) CT from May 11, 2021. (**C**) CT from 7 February 2022.

**Figure 3 cancers-14-02561-f003:**
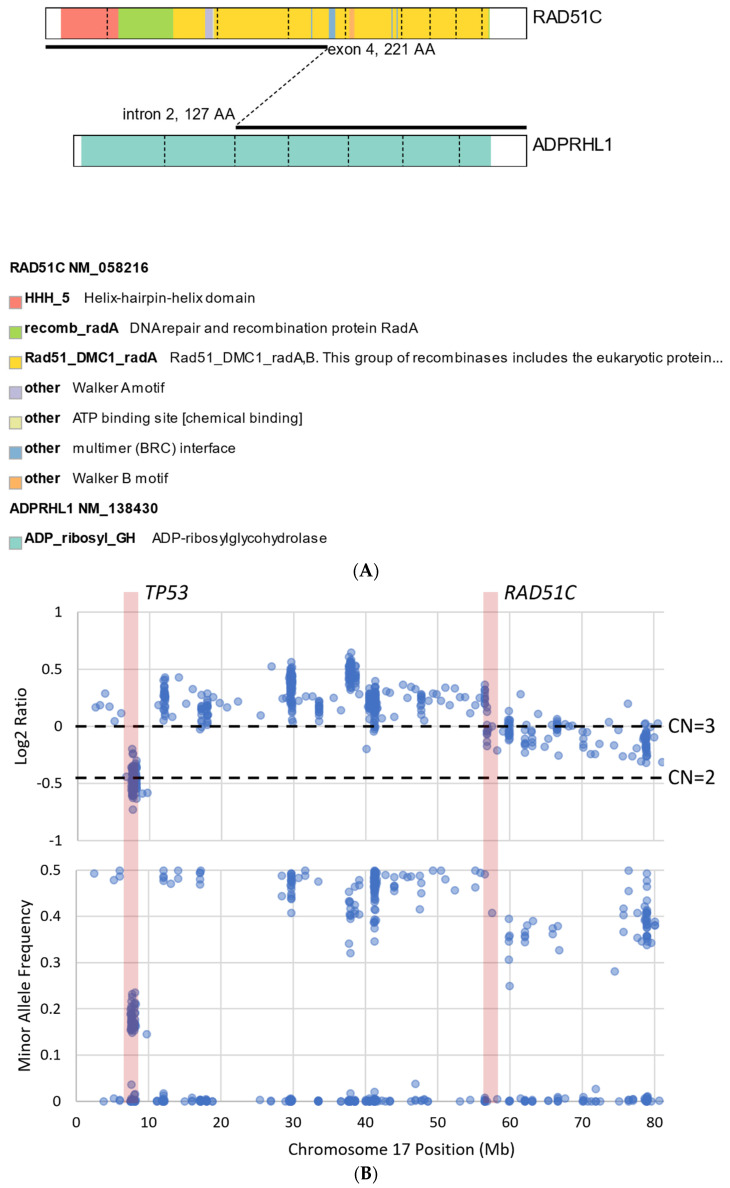
Graphical illustration of *RAD*51C mutation from the same patient presented in Figure 2. (**A**) Translocation t(13;17) with chromosome 17 breakpoint identified in exon 4 of *RAD*51C. (**B**) Examination of chromosome 17 copy number data shows a copy number transition accompanying LOH with a breakpoint at the *RAD*51C locus, suggesting likely loss of the second allele.

**Table 1 cancers-14-02561-t001:** Genes involved in HR, their frequencies in BTC, and a summary of evidence in support of sensitivity to PARPi when mutated.

Gene	Role/Mechanism in HR	Frequency in BTC [22,23]	Highest Level of Evidence Isolating Gene Mutation to PARPi Sensitivity
ARID1A	Mostly exerts a role in NHEJ mechanisms. It is a member of the SWI/SNF complex, which is involved in chromatin remodeling and essential for DNA repair [31,32].	13%	Cell lines: Loss of ARID1A shows sensitivity to PARPi [33]. Clinical: Retrospective data suggest that ARID1A loss may result in PARPi resistance in breast and ovarian cancer [34].
ATM	Functions upstream in the HR pathway. It helps phosphorylate and activate downstream HR effectors, such as BAP1, CHK2, and WRN [35].	5.7%	Animal model: ATM loss in prostate cancer leads to sensitivity to PARPi and ATR inhibition [36,37]. Cell lines: ATM loss in colorectal cell lines demonstrate sensitivity to Olaparib [38]. Clinical: Phase 3 trial of olaparib plus paclitaxel in advanced gastric cancer did not show improved mOS in the olaparib arm for patients deficient in ATM (via immunostaining) [39]. ATM inhibitors are being tested in early phase trials.
ATR	Regulates cell cycle checkpoint and mitotic entry. It phosphorylates and activates downstream CHK1 and WRN [40].	5.1%	Xenograft: PARPi with ATR inhibitor helps overcomes platinum resistance in ovarian cancer models [41].
ATRX	Involved in chromatin remodeling and is part of the SWI/SNF family. Operates downstream of RAD51. It is necessary for DNA repair synthesis and formation of sister chromatid exchanges at DSBs [42].	4%	Cell line: ATRX knockout cells were susceptible to PARPi (and ATR inhibition) [43].
BAP1	Involved in chromatin modulation and transcriptional regulation. It localizes in the endoplasmic reticulum, where it binds, deubiquitylates, and stabilizes IP3R3, modulating calcium release from the endoplasmic reticulum and apoptosis [44]. In HR, it regulates and recruits key downstream effectors, including p53, BRCA1, and RAD51. It is phosphorylated by ATM [45].	7.4%	Case report: Patient with refractory metastatic CCA with novel BAP1 mutation (splice site c.581-17_585del22) had a good, prolonged response to olaparib (>11 months) [46]. Clinical: Rucaparib in patients with BAP1-deficient (by immunostaining) or BRCA1-deficient recurrent mesothelioma showed early signs of efficacy (disease control rate at 12 weeks was 58%) [47].
BARD1	BRCA1-associated RING Domain Protein 1 upon genotoxic stress, BARD1 serves as a BRCA1 nuclear chaperone that promotes the formation and retention of BRCA1 foci, and these foci are colocalized with DNA repair effectors such as BRCA2 and RAD51 [48].	2.5%	Cell line: Colon cancer cells with BARD1 loss of function are more aggressive but sensitive to PARPi [49].
BLM	Unwinds dsDNA and regulates RAD51 foci formation. It is part of the BTR complex [35].	1.9%	Cell line: in NSCLC cells, BLM inhibitor sensitized sells to PARPi-medicated radiosensitization [50]. Another study demonstrated the synergy of BLM helicase inhibitor with PARPi in colon cancer cells [51].
BRCA1/2	BRCA1 promotes HR over NHEJ by directly interacting with PALB2 and recruiting BRCA2/RAD51 to DSBs.	1.9% for BRAC1, 4.4% for BRAC2	Clinical: Very limited data in CCA; however, there is robust data in phase III trials with PARPi for other cancers of the prostate, breast, ovarian, and pancreas.
CHK1/2	CHK1 is mostly phosphorylated by ATR; CHK2 by ATM. Regulates cell cycle checkpoint and DNA fork stabilization [52].	1.0% for CHK1, 1.9% for CHK2	Cell line: CHK1 knockout gastric cancer cell line was suspectable to PARPi. Synergy was shown between PARPi and CHK1 inhibitor [53].
FANC	A group of proteins forming the Fanconi Anemia core complex, which participates in HR by attracting HR effectors to the DSB site. FANCD1 gene is otherwise known as BRCA2. [54] FANCA is the most commonly altered FANC gene.	2.5%,	Clinical: Very limited data in CCA; however, the data for use of PARPi in FANCD1/BRCA2 mutation in other cancers is robust. In the TRITON2 study (rucaparib in prostate cancer), of 4 patients with a FANCA alteration, one patient with a monoallelic truncating alteration had a complete response [55].
NBN or NBS1	Recognizes and localizes to DSB sites. It recruits ATM and ATR. It is part of the MRN complex (MRE11-RAD50-NBS1) [56].	1.4%	Cell line: Dual disruption of MRN complex and PARP inhibition showed synergy in BRCA-proficient head and neck cancer cells [57].
PALB2	Localizes with BRCA2 and recruits RAD51 to the DBS site [35].	1.9%	Clinical: Phase II trial for metastatic breast cancer with HR mutations showed an ORR of 82% for germline PALB2 mutations when treated with olaparib [58]. A phase II study of maintenance rucaparib in advanced pancreatic cancer showed an ORR of 50% in germline PALB2 mutation.
RAD50	A critical part of the MRN complex (MRE11-RAD50-NBS1). Recognizes, localizes, and recruits HR effectors to DSB sites [56].	1.8%	Cell line: RAD50 depletion using siRNA in cancer cells showed increased platinum sensitivity [59]. Knockout of RAD50 in ovarian cancer cell lines yielded better responses to olaparib and rucaparib [60]. Clinical: A retrospective study of BRCA wild-type ovarian cancer showed somatic copy number deletion of RAD50 (by mRNA testing) led to a higher genome-wide mutation rate and increased sensitivity to olaparib and rucaparib [60].
RAD51 and paralogs	Physical interaction between BRCA2 and RAD51 is essential for error-free DSB repair [61]. BRCA2 is required for the localization of RAD51 to sites of DNA damage, where RAD51 forms the nucleoprotein filament required for recombination. The foci of the RAD51 protein are apparent in the nucleus after certain forms of DNA damage, and these likely represent sites of repair by HR. BRCA2-deficient cells do not form RAD51 foci in response to DNA damage. [62,63].	4.6% for RAD51B, 1.8% for *RAD*51C	Cell line: Silencing of RAD51 expression increases sensitivity to PARPi [31]. Clinical: In breast cancer patients, immunostaining of RAD51 nuclear foci showed that increased expression of RAD51 nuclear foci correlated with PARPi resistance in BRCA mutated tumors [64]. In breast cancer specimens, the presence of RAD51 foci by immunostaining predicted resistance to DNA-damaging therapy [65].
RAD52	Binds single-stranded DNA and plays a key part in single-strand annealing and HRR of DSBs. In mammals, RAD52 is diminished in HRR compared to other proteins, including BRCA1/2 but may compensate for BRCA1/2 deficiencies. In checkpoint-deficient cells, it facilitates break-induced replication (BIR) [66,67,68].One of its most prominent functions is in the formation of a Rad51–Rad52–Rad59 complex [67].		Cell line: Dual suppression of RAD52 and PARP1 via inhibitors demonstrate a synergistic effect in BRCA1/2-deficient cells in vitro and in vivo [69].
WRN	Recruited to the sites of collapsed replication forks and is phosphorylated at multiple Ser/Thr sites by ATM, ATR, and CDK1 kinases. WRN binding to perturbed replication forks not only stabilizes RAD51 and the replication fork but also prevents excessive nuclease activities of MRE11 and/or EXO1 [70,71]	4.9%	Cell line: Combining siRNA-mediated silencing of WRN in head and neck squamous cell carcinoma augmented sensitivity to cisplatin [72].

## Data Availability

This article is intented to be a review article; no investigational data was reported.

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
