# Peer review of "Homologous Recombination Repair in Biliary Tract Cancers: A Prime Target for PARP Inhibition?"

_cancers, 2022, doi:10.3390/cancers14102561_

Round 1

Reviewer 1 Report

In this report, the author review how HR repair inhibition in biliary tract cancers affects treatment and cancer progression. They discuss PARPi inhibitors primarily but also cases with BRCA mutations (BRACAness). The review provides some insight, but the discussion of repair pathways is often simplistic and sometimes wrong. For example, important genes such as RAD52 and KU70/80 are not even mentioned. Additionally, the authors fail to realize that RAD51 loading unto resected ssDNA occurs by two parallel pathways: BRCA1/2 or RAD52 despite ample evidence. In fact, targeting RAD52 in BRCA2 mutants is actively pursued as the best therapy. The authors should also read more specific literature on how DNA SSBs and DSBs are repaired and cite appropriate papers. As it stands now, this paper requires a thorough re-write and re-analysis.

Comments

  1. The introduction on DNA damage repair should be thoroughly expanded. As it is written now is either simplistic or wrong. For example, the sentence at line 86 “Whereas single-stranded breaks (SSB) are identified and repaired by pathways such as base excision repair (BER), nucleotide excision repair (NER) or mismatch mediated repair (MMR)” is misleading if not wrong. BER has some role in single strand break repair but NER and MMR almost none. In fact, most single strand breaks are repaired by HR subpathways such as Break induced replication. For example, a single strand break is converted to a one ended DNA double strand break which is repaired by BIR. Reference 25 is not the appropriate reference here because they are also oversimplifying the problem. Please review papers on DNA damage repair and cite them here.
  2. A key gene left out from Table 1 is RAD52. Recent evidence shows that RAD52 is critical for HR particularly in the absence of BRCA1/2. In fact, RAD52 is targeted for inhibition with small molecule inhibitors because of its synthetic lethality with BRCA1/2 and PALB 2. The “restoration of HRR” pathway discussed at line 329 also has to do with RAD52 taking over in BRCA1/2 mutant cells. The role of RAD52 should be absolutely included in this discussion. The entire MRN complex and the KU70/80 heterodimer should also be discussed thoroughly as they are break sensors. CtIP as well because mutations in these genes are found in biliary tracts (PMID: 33339169, and see references on ku70 in there).
  3. This sentence “However, with some exceptions, including BRCA1/2 mutations, many of these defects and their impact on DDR are not yet well-understood” is again oversimplified. In fact, I would argue that we know less about what BRCA1 does than the MRN complex or RAD51. This sentence which the authors state correctly captures how we don’t know that much about BRCA1/2 “is evident across many cancers that are BRCA-wild type or have BRCA mutations of unclear significance.”
  4. The discussion on ARID1A is again simplistic. This gene has major functions in transcription in addition to DNA damage repair. This section should include a discussion on how mutations in ARID1A leads to global transcription deregulation in cancer cells. Also, where is the evidence for this sentence: “ARID1A and TP53 is often found to be co-mutated in CCA”? In fact, if you look at figure 5 of reference 41 it shows quite the opposite. Note that the TP53 bars rarely overlap with ARID1A in both cohorts tested. Also, if you read the introduction paragraph from this paper (PubMed ID34941867) you will find that co-occurring TP53 and ARID1A mutations are rare in cancers. I suggest the authors read this sentence from reference 34941867 “Since then, numerous reports have observed ARID1A and TP53 alterations co-occur less frequently than expected by chance in other human cancer types, including gastric, breast, and esophageal [23–26]”, and check out references 23-26 provided.
  5. IDH1 discussion. Please mention that IDH1 mutation must be heterozygous for the 2-HG production and primarily because of the driver R132H mutation.

Author Response

In this report, the author review how HR repair inhibition in biliary tract cancers affects treatment and cancer progression. They discuss PARPi inhibitors primarily but also cases with BRCA mutations (BRCAness). The review provides some insight, but the discussion of repair pathways is often simplistic and sometimes wrong. For example, important genes such as RAD52 and KU70/80 are not even mentioned. Additionally, the authors fail to realize that RAD51 loading unto resected ssDNA occurs by two parallel pathways: BRCA1/2 or RAD52 despite ample evidence. In fact, targeting RAD52 in BRCA2 mutants is actively pursued as the best therapy. The authors should also read more specific literature on how DNA SSBs and DSBs are repaired and cite appropriate papers. As it stands now, this paper requires a thorough re-write and re-analysis.

Comments

  1. The introduction on DNA damage repair should be thoroughly expanded. As it is written now is either simplistic or wrong. For example, the sentence at line 86 “Whereas single-stranded breaks (SSB) are identified and repaired by pathways such as base excision repair (BER), nucleotide excision repair (NER) or mismatch mediated repair (MMR)” is misleading if not wrong. BER has some role in single strand break repair but NER and MMR almost none. In fact, most single strand breaks are repaired by HR subpathways such as Break induced replication. For example, a single strand break is converted to a one ended DNA double strand break which is repaired by BIR. Reference 25 is not the appropriate reference here because they are also oversimplifying the problem. Please review papers on DNA damage repair and cite them here.

Thank you for this input. Although we feel that the intent our paper is not to comprehensively review the different mechanisms of DDR, we do feel that it important not to give oversimplified misleading information. As such, we have amended our section for a brief overview of DDR as suggested.

  1. A key gene left out from Table 1 is RAD52. Recent evidence shows that RAD52 is critical for HR particularly in the absence of BRCA1/2. In fact, RAD52 is targeted for inhibition with small molecule inhibitors because of its synthetic lethality with BRCA1/2 and PALB 2. The “restoration of HRR” pathway discussed at line 329 also has to do with RAD52 taking over in BRCA1/2 mutant cells. The role of RAD52 should be absolutely included in this discussion. The entire MRN complex and the KU70/80 heterodimer should also be discussed thoroughly as they are break sensors. CtIP as well because mutations in these genes are found in biliary tracts (PMID: 33339169 and see references on ku70 in there).

We have added a brief paragraph on RAD52, as well as noting its potential role as a compensatory mechanism in BRCA1/2 deficient cells. We also note in our revision that KU70/80 heterodimer compete with RAD52 in biding to DSB and could confer resistance to PARPi; however, these genes have not been reported in BTC. We also hope to focus the attention on HRR mechanisms (acknowledging that other mechanism also interplay in DDR.) We have added RAD52 into our table.

  1. Cell Rep. 2018 June 12: 23(11): 312
  2. Choi, YE, Cancer Report 2016; 14: 429-439

  1. This sentence “However, with some exceptions, including BRCA1/2 mutations, many of these defects and their impact on DDR are not yet well-understood” is again oversimplified. In fact, I would argue that we know less about what BRCA1 does than the MRN complex or RAD51. This sentence which the authors state correctly captures how we don’t know that much about BRCA1/2 “is evident across many cancers that are BRCA-wild type or have BRCA mutations of unclear significance.”

We have rephrased the former sentence: “However, the impact of many genes in the DDR pathway are not yet well-understood” to avoid being misleading

  1. The discussion on ARID1A is again simplistic. This gene has major functions in transcription in addition to DNA damage repair. This section should include a discussion on how mutations in ARID1A leads to global transcription deregulation in cancer cells. Also, where is the evidence for this sentence: “ARID1A and TP53 is often found to be co-mutated in CCA”? In fact, if you look at figure 5 of reference 41 it shows quite the opposite. Note that the TP53 bars rarely overlap with ARID1A in both cohorts tested. Also, if you read the introduction paragraph from this paper (PubMed ID34941867) you will find that co-occurring TP53 and ARID1A mutations are rare in cancers. I suggest the authors read this sentence from reference 34941867 “Since then, numerous reports have observed ARID1A and TP53 alterations co-occur less frequently than expected by chance in other human cancer types, including gastric, breast, and esophageal [23–26]”, and check out references 23-26 provided.

Thank you for this input. We have elaborated on the function of ARID1A and regarding its exclusivity with TP53.

  1. IDH1 discussion. Please mention that IDH1 mutation must be heterozygous for the 2-HG production and primarily because of the driver R132H mutation.

Thank you, we have included this comment in our revised review along with citations.

Reviewer 2 Report

Dear Editor, thank you so much for inviting me to revise this manuscript about biliary tract cancer.

The overall limited survival benefit provided by systemic therapies in this setting, with most patients reporting a survival rate of less than a year from the moment of diagnosis, has led to notable efforts toward the identification of novel targets and agents that could modify the natural history of these aggressive hepatobiliary malignancies. In fact, the massive use of next-generation sequencing (NGS) has led to the identification of previously unknown molecular features of CCA, including the presence of specific genetic aberrations that have been suggested to be distinctive features of iCCA and eCCA. Among these druggable alterations, fibroblast growth factor receptor (FGFR)2 gene fusions and rearrangements, isocitrate dehydrogenase-1 (IDH-1) mutations, and BRAF mutations have been widely described in CCA patients, reporting important differences between iCCA and eCCA.

Based on these premises, the paper addresses a timely topic.

The manuscript is quite well written and organized.

Tables are comprehensive and clear.

The introduction explains in a clear and coherent manner the background of this study.

We suggest the following modifications:

1. Figure 1 is not useful and could be removed.

2. Although the authors correctly included important papers in this setting, we believe a couple of papers should be cited (PMID: 32994319 ; PMID: 33611090  ), only for a matter of consistency. 

3. In addition, we believe some issues deserve further discussion. In everyday clinical practice, we know that the pathologic confirmation of diagnosis is necessary before any non-surgical treatment and can be challenging in BTC, particularly in patients affected by primary sclerosing cholangitis and biliary strictures. In fact, decisions to undertake biopsies should follow a multidisciplinary discussion, especially in potentially resectable tumors. Moreover, endoscopic imaging and tissue sampling are useful but, sadly, biopsy samples are often inadequate for molecular profiling, and in addition, tissue sampling has reported high specificity but low sensitivity in diagnosis of malignant biliary strictures. Finally, the highly desmoplastic nature of BTC limits the accuracy of cytological and pathological approaches.

On the basis of these premises, in this scenario, it is urgent to develop new strategies in order to anticipate the diagnosis identifying BTC at an early, resectable stage, and to obtain sufficient material with which to perform genomic analysis. Among these strategies, liquid biopsy has received growing attention over the years, given the promising applications in cancer patients. More specifically, several studies have shown the potential role of liquid biopsy, and the authors should discuss this point, also reporting recent studies in this setting (doi: 10.3390/cells9030721; doi: 10.21873/cgp.20203).

We believe that major revisions are needed. The main strengths of this paper are that it addresses an interesting and very timely question and provides clear answers, with some limitations. We suggest the addition of some references for a matter of consistency. Moreover, the authors should better clarify some points and should add some details and studies, as suggested.

Author Response

Review 2:

Comments and Suggestions for Authors

Dear Editor, thank you so much for inviting me to revise this manuscript about biliary tract cancer.

The overall limited survival benefit provided by systemic therapies in this setting, with most patients reporting a survival rate of less than a year from the moment of diagnosis, has led to notable efforts toward the identification of novel targets and agents that could modify the natural history of these aggressive hepatobiliary malignancies. In fact, the massive use of next-generation sequencing (NGS) has led to the identification of previously unknown molecular features of CCA, including the presence of specific genetic aberrations that have been suggested to be distinctive features of iCCA and eCCA. Among these druggable alterations, fibroblast growth factor receptor (FGFR)2 gene fusions and rearrangements, isocitrate dehydrogenase-1 (IDH-1) mutations, and BRAF mutations have been widely described in CCA patients, reporting important differences between iCCA and eCCA.

Based on these premises, the paper addresses a timely topic.

The manuscript is quite well written and organized.

Tables are comprehensive and clear.

The introduction explains in a clear and coherent manner the background of this study.

We suggest the following modifications:

  1.     Figure 1 is not useful and could be removed.

The figure was requested by the journal

  1.     Although the authors correctly included important papers in this setting, we believe a couple of papers should be cited (PMID: 32994319 ; PMID: 33611090  ), only for a matter of consistency.

We have included these citations into the revision.

  1.     In addition, we believe some issues deserve further discussion. In everyday clinical practice, we know that the pathologic confirmation of diagnosis is necessary before any non-surgical treatment and can be challenging in BTC, particularly in patients affected by primary sclerosing cholangitis and biliary strictures. In fact, decisions to undertake biopsies should follow a multidisciplinary discussion, especially in potentially resectable tumors. Moreover, endoscopic imaging and tissue sampling are useful but, sadly, biopsy samples are often inadequate for molecular profiling, and in addition, tissue sampling has reported high specificity but low sensitivity in diagnosis of malignant biliary strictures. Finally, the highly desmoplastic nature of BTC limits the accuracy of cytological and pathological approaches.

Thank you for these comments. We would like to focus on the topic of DDR and PARPi use in BTC. However, we recognize the limitations that you have described, which are central to the overall treatment of BTC. As such, we have made brief acknowledgement of these issues in our discussion.

On the basis of these premises, in this scenario, it is urgent to develop new strategies in order to anticipate the diagnosis identifying BTC at an early, resectable stage, and to obtain sufficient material with which to perform genomic analysis. Among these strategies, liquid biopsy has received growing attention over the years, given the promising applications in cancer patients. More specifically, several studies have shown the potential role of liquid biopsy, and the authors should discuss this point, also reporting recent studies in this setting (doi: 10.3390/cells9030721; doi: 10.21873/cgp.20203).

Thank you for your input. We have briefly summarized this into our conclusion in our revision.

Round 2

Reviewer 1 Report

The authors have made significant changes. This reviewer is satisfied. 

Reviewer 2 Report

Acceptance.